



# CLIMADA v1.4.1: Towards a globally consistent adaptation options appraisal tool

David N. Bresch[1,2], Gabriela Aznar-Siguan[2]

[1]Institute for Environmental Decisions, ETH Zurich, Zurich, Switzerland
[2]Federal Office of Meteorology and Climatology MeteoSwiss, Zurich, Switzerland

*Correspondence to*: David N. Bresch (dbresch@ethz.ch)

**Abstract.**

Climate change is a fact and adaptation to a changing environment therefore a necessity. Adaptation is ultimately local, yet similar challenges pose themselves to decision-makers all across the globe and on all levels. The Economics of Climate
Adaptation (ECA) methodology established an economic framework to fully integrate risk and reward perspectives of different stakeholders, underpinned by the CLIMADA impact modelling platform. We present an extension of the latter to appraise adaption options in a consistent fashion in order to provide decision-makers from the local to the global level with the necessary facts to identify the most effective instruments to meet the adaptation challenge. We apply the open-source methodology and its Python implementation to a case study in the Caribbean, which allows to prioritize a small basked of adaptation options,
namely green and grey infrastructure options as well as behavioural measures, and permits inter-island comparisons. In Anguilla, for example, mangroves avert simulated damages more than 4 times the cost estimated for restoration, while enforcement of building codes shows to be effective in the Turks and Caicos islands. For all islands, cost-effective measures reduce the cost of risk transfer, which covers damage of high impact events that cannot be cost-effectively prevented by other measures. This extended version of the CLIMADA platform has been designed to enable risk assessment and options appraisal
in a modular form and occasionally bespoke fashion yet with high reusability of common functionalities to foster usage of the platform in interdisciplinary studies and international collaboration.

## 1 Introduction

Climate change is one of the defining challenges of mankind in the present century. Even if we would stop  global greenhouse gas emissions today, we are bound to a significant level of warming and concomitant change (IPCC, 2014). Adaptation to a
changing environment therefore, shaped not only by changes in climate, but also societal stressors is and will remain a key priority today and in future. Adaptation is ultimately local, yet similar challenges pose themselves to decision-makers all across the globe and on all levels – from multinational organizations (Berkhout, 2012), sovereign and sub-sovereign states, cities, companies, and down to the local community (Webler et al., 2016). They all benefit from consistent methodologies providing the facts to identify the most effective instruments to meet the adaptation challenge despite constraints such as insufficient



local resources, capacities and authority. Such an approach needs to combine impact (Burton et al., 2002; Füssel and Klein, 2006) with vulnerability (Fünfgeld and Mcevoy, 2011; Preston et al., 2011) assessments to strengthen societal resilience through co-generation of adaptation knowledge (Muccione et al., 2019), dissemination of information (Moser, 2014, 2017) and enabling access to funding (Adger, 2006; Eakin and Lemos, 2006; Smit and Wandel, 2006; Yohe and Tol, 2002). In this

spirit, the Economics of Climate Adaptation (ECA) methodology established an economic framework to fully integrate risk and reward perspectives of different stakeholders (Bresch, 2016; Bresch and ECA working group, 2009; Bresch and Schraft, 2011; Souvignet et al., 2016) to foster climate-resilient development (Watkiss and Hunt, 2016). ECA can be applied on different levels and granularity, combining elements of top-down and bottom-up approaches (e.g. Dessai et al., 2005), both used in the policy process (Kates and Wilbanks, 2003; Mc Kenzie Hedger et al., 2006), as it provides the fact base to build an

adaptation strategy that is robust against a wide range of plausible climate and societal change futures (Lempert and Schlesinger, 2000; Wilby and Dessai, 2010). ECA starts with a comprehensive cost benefit analysis (CBA), where benefit does not need to be expressed in monetary units, but equally well e.g. in lives saved, as illustrated by many case studies (Bresch and ECA working group, 2009; Wieneke and Bresch, 2016). Such CBA forms the basis for a wider Multi-Criteria Analysis (MCA, e.g. Haque (2016)) to integrate aspects such as specific risk appetite and further locally determined context (Brown et

al., 2011; Dessai and Hulme, 2004; Preston and Stafford-Smith, 2009; Truong et al., 2016) and criteria (e.g. in NAPAs, UNFCCC, 2011) as well as additional perspectives (Radhakrishnan et al., 2017), including – also indigenous (Kelman et al., 2012) – knowledge with respect to feasibility. This will allow to (re-)prioritize measures to constitute an adaptation roadmap as a basis for adaptation funding discussions on all levels, from local to global, including e.g. the Green Climate Fund (GFC). The ECA method is underpinned by the CLIMADA platform (Aznar-Siguan and Bresch, 2019b) which does allow for globally

consistent (c.f. Ward et al. (2020)) yet high-resolution modelling of socioeconomic impacts of weather extremes following a fully probabilistic event-based approach. Impacts are assessed today, as well as in future, subject to the increase due to economic development, and the further incremental increase of risk due to climate change.

Building on the risk assessment already implemented (Aznar-Siguan and Bresch, 2019b), the present paper describes the concept of options appraisal by estimating the expected change in socioeconomic impact over time. This allows risk reduction

benefit to be compared to the implementation cost of options, covered in Section 2, where the object-oriented design of the Python implementation is also documented. Section 3 provides an exemplary case study application and Section 4 concludes with discussion and outlook. This extended version of the CLIMADA platform has been designed to enable risk assessment and options appraisal in a modular form and occasionally bespoke fashion (Hinkel and Bisaro, 2016) yet with high reusability of common functionalities to foster usage in interdisciplinary studies (Souvignet et al., 2016) and international collaboration.





## 2 Framework Concept and Design

### 2.1 Concept

Framing the climate adaptation challenge in terms of risk allows to treat adaptation measures as ways to reduce natural hazard risk both today and in future. Starting from a calibrated impact model to assess current risk (as e.g. in Aznar-Siguan and Bresch 2019), the drivers of risk are first modified to implement the effect of changes in hazard (e.g. climate-driven) and exposure (development-driven) over time, likely in a scenario fashion (c.f. Serrao-Neumann and Low Choy (2018)). Changes in vulnerability could easily be taken into account, too, but pertinent information does usually not exist. Adaptation measures are implemented through modification of the impact function (e.g. better building codes leading to lower building damages), possibly also exposure or hazard, or a combination thereof. Risk metrics both for today as well as for future years can thus be calculated with and without any such measure. The (net present value of) the difference of the risk metrics computed with and without the implementation of the measure constitutes the measure's benefit. Together with estimates of implementation (capital expenditures, CAPEX) and maintenance (operations expenditures, OPEX) costs (or payment streams thereof, c.f. Samuelson (1937)), cost-benefit ratios can be calculated. Please note that risk metrics need not be monetary, hence discounting or more general questions of time preference (Frederick et al., 2002) of the benefits might not be directly applicable (e.g. for risk metrics number of people displaced or lives lost) and costs could also be specified in non-monetary units. Please note further that climate scenarios and development pathways are usually employed to assess future risk, hence such cost-benefit considerations are contingent on the scenarios and pathways chosen for analysis. But as it will be shown, very much in the spirit of Wilby and Dessai (2010), (baskets of) adaptation measures can thus be tested for robustness (Dittrich et al., 2016) under different combinations of scenarios and pathways (as well as other key parameters, such as time-dependent discount rates).

### 2.2 Implementation

The software architecture defined in Aznar-Siguan and Bresch (2019) has been extended to include the classes which handle adaptation measures (*Measure* and *MeasureSet*), discount rates (*DiscRates*) and the cost/benefit analysis (*CostBenefit*), as shown in Figure 1. Note that other evaluation approaches, such as real options (Hino and Hall, 2017) or multi-criteria analysis (Haque, 2016), could be implemented in a similar fashion, in close correspondence to the cost/benefit as shown here.





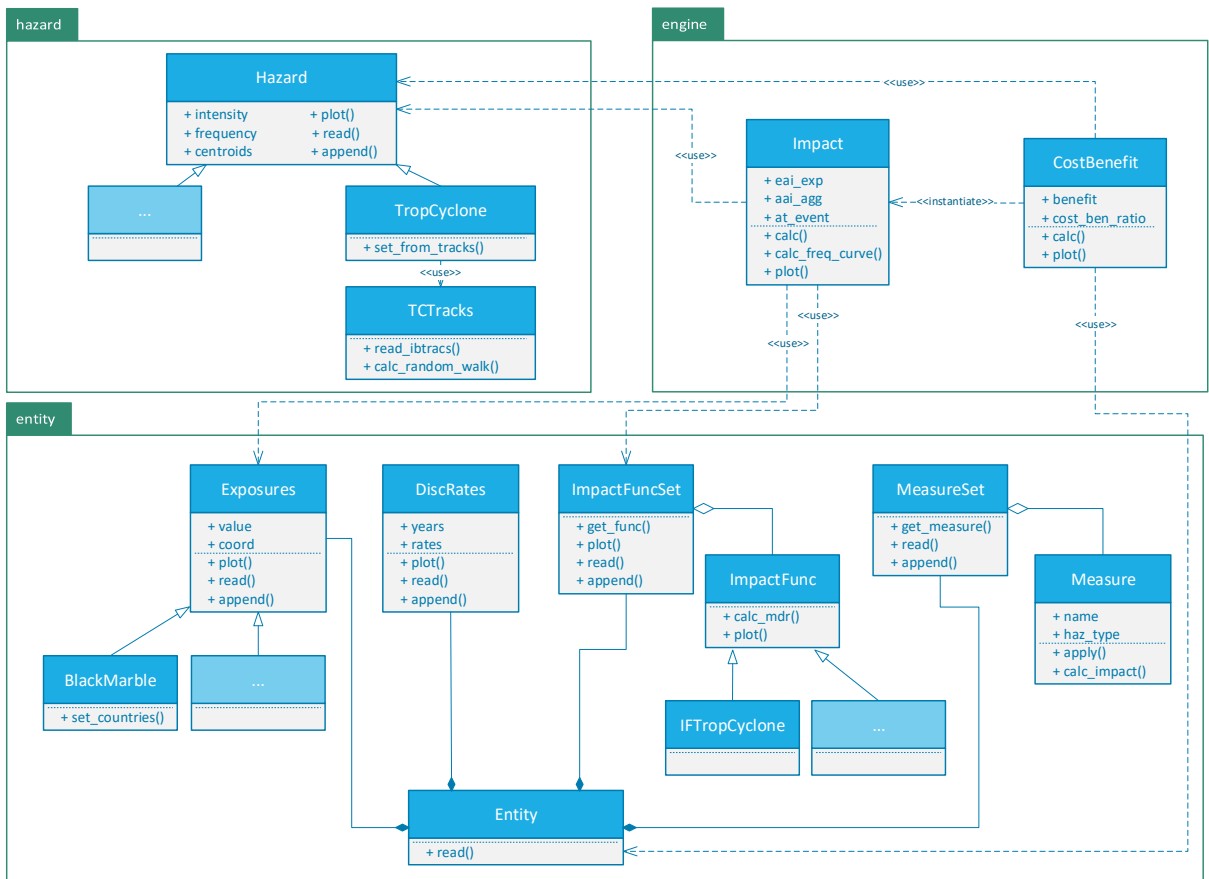

**Figure 1: Simplified architecture of CLIMADA including classes for adaptation measures, discount rates and cost-benefit considerations. An extension of Aznar-Siguan and Bresch (2019).**

### 2.2.1 Adaptation Measures

An adaptation measure in CLIMADA is a parametrization of a risk-reducing measure which modifies either the impact function, the exposure or the hazard, or a combination of any of them, or even the resulting impact. A measure is defined in the *Measure* class and is uniquely identified by its name and the hazard type it is acting on. Its parametrization is implemented via attributes. *exp_region_id* sets the physical boundaries of the measure, where exposures and hazards outside the defined regions are not modified by the measure; *hazard_set* and *hazard_freq_cutoff* change a given *Hazard* instance. The first replaces

the hazard by a new *Hazard* instance which allows for the flexibility to introduce any desired protection distribution, spatially and in frequency of occurrence (e.g. a flood hazard event set built with higher flood protection in place). *hazard_freq_cutoff* defines a frequency cut-off which sets impacts at higher frequency to zero and can thus be used to model a seawall, for example, which avoids all impacts with a frequency higher than *hazard_freq_cutoff* within its protected region defined by *exp_region_id*. The exposures are modified through the parameters *exposures_set* and *imp_fun_map*. *exposures_set* replaces the *Exposure*

instance and *imp_fun_map* changes the selected impact functions assigned to each exposure for others. *exposures_set* provides





more freedom to define changes on the exposure, such as changes in the assets distribution through modified spatial planning. Implementing a building code for a specific construction type could be modelled with a new impact function and relating it to the former one through the *imp_fun_map* attribute.

Even if new impact functions can be easily introduced, the following attributes allow to perform linear transformations to given impact functions: *hazard_inten_imp* transforms the abscissae (e.g. implementing elevation of homes in the case of flood) while *mdd_impact* and *paa_impact* transform, respectively, the Mean Damage Degree and the Percentage of Affected Assets (e.g. to reflect an improved building code).

Finally, a classical risk transfer option can be defined setting a deductible (or attachment point) and a cover. Deductible and cover are considered for the resulting damage of each event. Damages greater than the *deductible* up to *cover* are carried by the insurer, hence lowering the damage burden for the insured. As the insurer incurs transaction and capital costs, the total cost of insurance is approximated by application of the multiplicative *risk_transf_cost_factor (>1, usually order of 1.5 to 2)* to the raw expected damage which is calculated for the insured layer (for details, see the case study below as well as e.g. Surminski et al. (2020)).

For measures other than risk transfer, their cost is provided by the user through the *cost* attribute. This should provide the Net Present Value (NPV) of the initial investment (capital expenditure, CAPEX) as well as the maintenance costs during the whole time range of implementation considered (operating expenditures, OPEX). The set of measures that are going to be compared in the cost-benefit analysis are gathered in the container *MeasureSet* represented in Figure 1.

### 2.2.2 Cost and Benefit

The *CostBenefit* class in Figure 1 computes the costs and benefits of implementing a set of adaptation measures through its *calc* method. There, the socioeconomic variables are provided by the *Entity* class, where the exposure to the hazard (*Exposures* instance), a set of impact functions representing the exposures vulnerability (*ImpactFuncSet* instance), a set of measures (*MeasureSet* instance) and the (if applicable time- and even measure-dependent) discount rates (*DiscRates* instance) to be applied over the time period of interest are gathered. The natural hazard is provided by a *Hazard* instance or a derivate class. Within the *calc* method the probabilities of impact resulting from the implementation of each measure are computed through the *Impact* class as explained in Aznar-Siguan and Bresch (2019) and compared to the risk when no measure is applied. The benefit of the measure is its averted impact in terms of a configurable "risk function". As a default, the average annual averted impact is used, but any risk function, such as any quantile or the averted impact for events with a specific return period (e.g. on in a hundred years) can be considered.

Scenarios of both future hazard as well as exposure at the end of the time period considered can be provided as well as follows: A second *Hazard* captures the changes of intensities and probability of occurrence and a second *Entity* contains the changed exposures, measures and eventually new impact functions (to account for change in building quality, for example). The probabilities of impact for each measure are computed at the beginning and at the end of the time range, and stored, respectively in the attributes *imp_meas_present* and *imp_meas_future*. The benefit is then computed as the NPV of the average annual



averted impact (or the configured risk function) using the discount rates of *DiscRates*. The values discounted in the years between the beginning and the end of the period are estimated by interpolation through the parameter *imp_time_depen*. This allows to set a linear (as by default) or either a concave (large increase at the beginning) or convex (increase mainly towards the end) change of risk over time.

The resulting benefits per measure are stored in the attribute *benefit*, while the cost/benefit ratio is stored per measure in the *cost_ben_ratio* attribute. These values are used by the method *plot_cost_benefit* where the benefit of each measure is represented against the corresponding benefit/cost ratio in an adaptation cost curve. *plot_event_view* provides a further understanding of the efficiency of the measures by showing the quantity of averted impact in events of selected return periods

10 at the end of the time range considered (see case study below for illustrations). If some of the measures are to be implemented simultaneously, the method *combine_measures* can be used to obtain an approximation of the combined averted impact. There, the benefits of the measures are aggregated at event level, avoiding double counting, and the risk function is applied afterwards. Furthermore, the *apply_risk_transfer* method allows to implement risk transfer on top of selected measures (after combination, if applied, to properly account for risk reduction and diversification effects).

## 15 3 Case study: Adaptation to hurricanes in the Antilles

Building on the risk assessment case study documented in Aznar-Siguan and Bresch (2019), we consider here the small Caribbean islands hit by hurricane Irma. The consequences of the 2017 Atlantic Hurricane season underscore the importance of investing in disaster risk reduction for resilience, enhancing disaster preparedness for effective response and the imperative of "Building Back Better" during recovery, rehabilitation and reconstruction (ECLAC, 2018).

CLIMADA has been used to quantify adaptation options before, see Bresch (2016), Bresch and ECA working group (2009), Bresch and Schraft (2011) and Souvignet et al. (2016). These assessments were performed following the Economics of Climate Adaptation (ECA) methodology within consortia where data provided by dedicated surveys and local experts fed the models. Please note that these studies were well embedded in local stakeholder consultation and co-design processes, especially

25 regarding both the scope as well as the set of adaptation measures considered. The analysis documented here aims at showing the versatility that CLIMADA offers to compare adaptation measures of different nature under different scenarios, but does not provide a fully comprehensive adaptation assessment in the sense of a full ECA (Souvignet et al., 2016). In order to keep the case study lean and illustrative, only openly available national indicators are used and uncertainties not explored in detail, even though CLIMADA is designed to do so. We first describe the approach for Anguilla in 3.1 and apply it later on all the

30 targeted islands in 3.2.





### 3.1 Adaptation in Anguilla

The analysis of Aznar-Siguan and Bresch (2019) concludes that the current Average Annual Impact (AAI) of hurricanes in Anguilla is 18±4 million current US dollars. The impacts were assessed in terms of physical damage on infrastructure whose value is proportional to its contribution to the national produced goods and services. Using these modelled assets, the generated

tropical cyclone events (historical and synthetic) and the impact function of the previous work, we define several adaptation measures as explained in 2.2.10, and quantify their cost and benefit in terms of physical protection following 2.2.2. The time frame considered for this study ranges from 2016 (hereafter referred to as 'current' time, establishing a risk baseline) until 2050.

#### 3.1.1    Adaptation measures definition

We consider measures of different nature, such as green and grey infrastructure options (Denjean et al., 2017) as well as behavioural measures. Ecosystem-based measures such as mangroves can provide substantial protection to properties, even relatively far away from them, against both storm surges and cyclonic wind (Das and Crépin, 2013; Reguero et al., 2018). The need on reforestation of mangroves in the Caribbean started in the 1980s, when large-scale conversion of mangroves for aquaculture and tourism infrastructure took place. Even if Anguilla has maintained its mangrove area relatively constant to 90

hectares (Food and Agriculture Organization of the United Nations, 2007), these can be further damaged by storms. We define an investment of 100'000 USD per hectare for restoration of Anguilla's mangroves (Lewis, 2001) and an annual maintenance cost of 200 USD per hectare that leads to a reduction of the impact function intensity of 0.74% on the coast and twice as much inland. The resulting impact functions (*mangrove_coast* and *mangrove_inland* respectively) are represented in Figure 2.

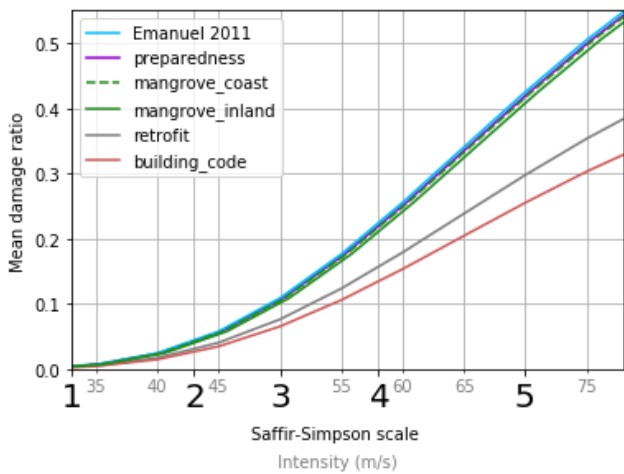

**Figure 2: Impact functions used in the definition of adaptation options.** *Emanuel 2011* **refers to the impact function used in the risk assessment, following (Emanuel, 2011), and the Saffir-Simpson hurricane wind scale is shown as reference (Schott et al., 2019).** *Preparedness* **is the impact function obtained once the behavioural measure named preparedness is implemented. Mangroves reforestations generate the impact functions** *mangrove_coast* **on a distance of up to 500 meters from the coast and** *mangrove_inland*



**in the rest of the island.** *Retrofit* **and** *building_code* **show an impact reduction of 30% and 40% respectively at every wind intensity with respect to** *Emanuel 2011*. **See Aznar-Siguan and Bresch (2019) for details about mean damage ratio (MDR).**

As grey options we consider retrofitting and the implementation of building codes. Retrofitting existing housing not only reduces damages due to natural disasters, but can also lead to a reduction in insurance costs (Surminski et al., 2020), potentially

increases the market value of a building, and may have co-benefits such as energy saving. Here we only consider the benefit caused just by physical protection, which we set to an idealized 30% of damage reduction at every wind intensity (see Figure 2). Retrofitting can cost anywhere between 1% and 20% of the value of the property (Ou-Yang et al., 2013; Triveno and Hausler, 2017) and can be efficiently subsidized by governments. For illustration purposes, we set a total cost of 10% of the retrofitted assets. The retrofit is performed progressively, having 10% of the assets value retrofitted in 2016 and achieving

90% in 2050.

Implementing building codes has similar benefits as retrofitting, but only in newly constructed buildings. Its success lies in its enforcement and subsequent inspection, especially in residential housing (Prevatt et al., 2010). To assess its benefits and costs, we consider an annual rate of urbanization in Anguilla of 0.9% (Central Intelligence Agency, 2019) and 40% reduction in the impact function (see Figure 2). To approximate the costs from the government to train construction workers and hire inspectors,

as well as the owners expenses, the cost is set to 5% of the annual newly built houses.

Preparing houses by protecting windows, roofs and clearing the exteriors helps to reduce damage. Such action can be explained and promoted through labels (Attems et al., 2020) leaflets or e.g. at the bottom of bills such as the electricity invoice. We name this measure *preparedness*, and approximate its cost per inhabitant as (i) the cost of communication of 1 USD plus (ii) an

annual 0.2 USD maintenance, and (iii) 100 USD as bulk expense for protection material. This measure is set to avoid all damages for events with return periods of up to 7 years and to reduce the vulnerability by modification of the curve along the wind intensity axis by 0.5% for events with higher return period (see impact function in Figure 2).

Finally, risk transfer is considered, being particularly suitable to manage risks of low frequency, high severity events. We

define an insurance layer with attachment point (or deductible) and cover proportional to the island's expected exceedance damages. The attachment is set to the 12 year events damages (approximately 32 m USD) and the cover to cater for events with up to 145 year return period (approximately 314 m USD). It comes at a cost of 1 m USD plus 2% of the cover amount (a simple proxy for transaction and capital costs) plus 1.5 times *(the risk_transf_cost_factor)* the expected damage in the insurance layer (more to illustrate the implementation in principle than to model a specific case).

### 3.1.2   Cost and benefit of adaptation measures under changing risk

Risk to tropical cyclones during the 35 years of implementation of the measures will change because of economic development (increased exposure and modified impact functions) and climate change (changing hazard). In order to assess the uncertainty of the future, CLIMADA compares different plausible future scenarios. We consider here a moderate scenario, where the



economic growth follows the trend of the previous years, a 2% annual increase, and the change in tropical cyclones follows a climate change stabilization scenario, the Representative Concentration Pathway (RCP) 4.5. We implement the consequent changes in intensity and frequency of tropical cyclones at 2050 following Knutson et al. (2015). Under this scenario the AAI is increased by 23 m USD, leading to a mean AAI of 41 m USD in 2050. Considering a linear change of risk during the 35

years and a discount rate of 2%, the Net Present Value (NPV) of the total expected damages is 723 m USD, 57% higher than without increase of risk due to economic development and (moderate) climate change, as illustrated in Figure 3. Almost one fourth of the damage increase in the moderate scenario is attributable to climate change, while the main increase is due to economic development. These changes in risk over time have a substantial impact on the cost-benefit analysis of the adaptation measures.

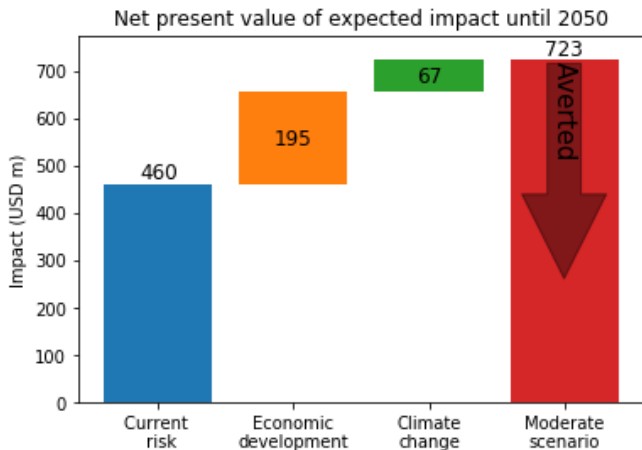

**Figure 3: Net Present Values (NPVs) of the expected tropical cyclone damages (average annual impact, AAI) in Anguilla by 2050.** *Current risk* **represents the NPV of the expected impacts from 2016 until 2050 if neither assets nor climate changes.** *Moderate scenario* **shows the NPV of expected impacts with moderate economic development and climate change following RCP 4.5. Separate contribution to the total expected impact are shown in columns** *Economic development* **and** *Climate change* **respectively. The arrow**
*Averted* **shows the quantity of impact that can be averted implementing the measures** *preparedness, mangrove, building_code* **and** *risk_transfer***, as explained below (Fig. 4).**

Figure 4 represents the NPV of the expected impact ("Total risk" tag) without changing risk (a) and with the moderate risk increase scenario (b). These amounts are compared to the total averted impact of each measure in the x-axis. Whilst with current risk the implementation of all measures could eventually lead to avert almost all of the expected impact, this is not

possible by 2050 any more, given the increase in risk both driven by economic growth and climate change (Fig 1.), despite the concomitant increase in averted impact of each measure. Nevertheless, the cost-effectiveness of all measures increases in the moderate scenario (and would even more so in a high change scenario). *Preparedness* and *mangrove* increase their benefit/cost ratio to values well above 1, 11.7 and 4.5 respectively, while the grey solutions effectiveness increase more moderately. *Retrofit* remains the measure averting most of the damage after *risk_transfer* but is still not cost-effective, reaching a benefit/cost ratio

of 0.88. *Building_code* averts more than 1 m USD than *mangrove* but its benefit/cost ratio stays bellow 2. The *risk_transfer*





option is not cost-efficient in the narrow sense, due to transaction and capital costs, but likely remains attractive to a risk averse agent nevertheless (e.g. Jullien et al., 1999).

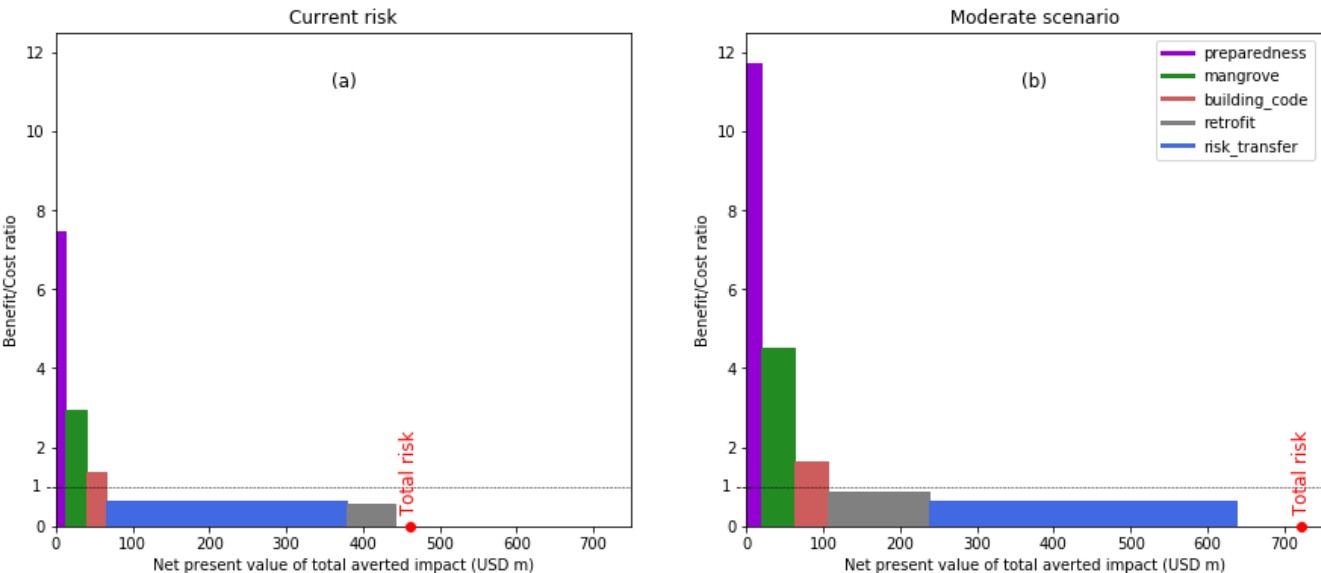

**Figure 4: Net present value (NPV) of each measures total benefit and benefit/cost ratio without a changing future (a) and with a**
5 **scenario of moderate change (b). "Total risk" indicates the NPV of the total damage expected if no measure is implemented (as in Fig 3. Above).**

How the measures perform is further represented in Figure 5. The expected exceedance damages for events of return periods 7, 10 and 40 are shown together with the amount of damage that every measure averts, both considering the current risk (a) and the moderate change scenario (b). *Preparedness* averts all the damages for events with return periods lower and equal than
10 7 years but its protection is minimal for less frequent events. This is not the case of *building_code,* which improves its performance with increasing return period events, the same way as *retrofit* does. The later just averts more damage than *building_code* because it is implemented more extensively. For events with 10 years return period the current risk scenario does not reach the attachment point of the *risk_transfer* (set at 12 years), while the increase in risk under the moderate scenario triggers risk transfer already more often than every 10 years in future. However, even by using *risk_transfer* solutions together
with all the other measures, 40-year events cannot be fully covered any more under the moderate change scenario by 2050.





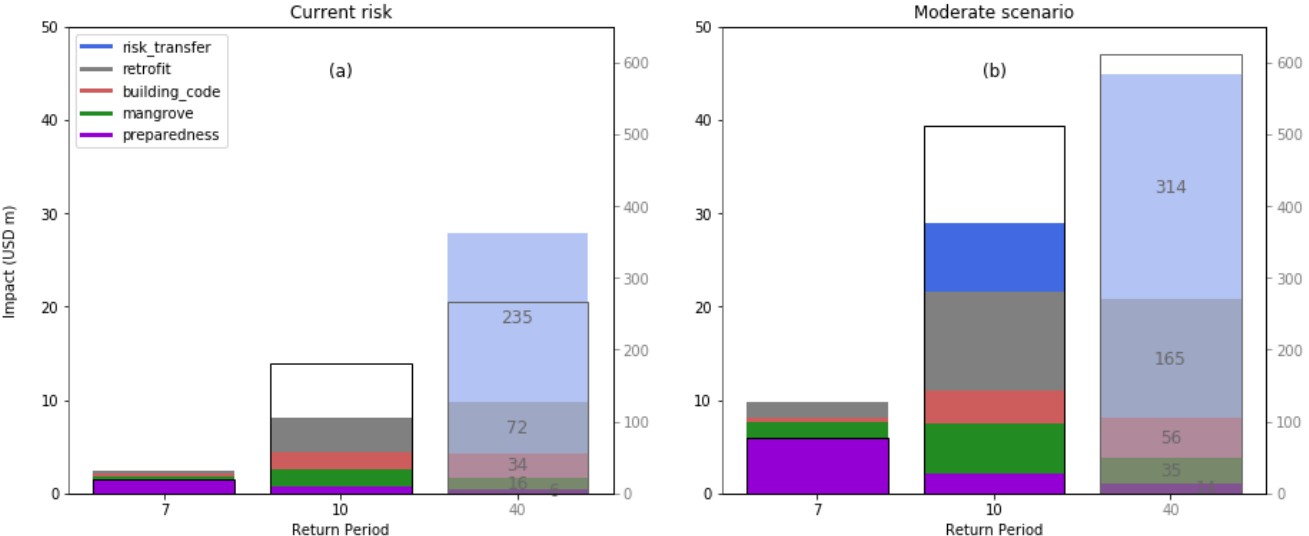

**Figure 5: Averted impact of each measure in 2050 for different return periods, without taking into account climate change nor economic growth (a) and with the moderate risk increase (b). The black bars show the expected exceedance damage at each return period and each coloured block indicates the amount averted by the corresponding measure. Note the different vertical scale (on the right) for 40 years return period. Shown are non-discounted values.**

### 3.1.3    Combining measures

Figure 4 represents the averted damages of each measure independently. Combining the three most cost-effective measures for Anguilla (*preparedness, mangrove* and *building_code*) in the moderate risk scenario only averts a total damage of 104 m USD, just slightly lower than their added benefits (as combining in CLIMADA avoids double-counting). Applying *risk_transfer* on top will further increase the averted damage to 469 m USD, 65% of all the expected damages (see averted damage in Figure 3). Even if *risk_transfer* alone was already averting 399 m USD, the difference of combining insurance with other adaptation solutions leads to a substantial reduction in cost for insurance. Implemented alone, *risk_transfer* costs 605 m USD, compared to 554 m USD when combined with *preparedness, mangrove* and *building_code*, leading to an improved insurance benefit/cost ratio of 0.79 instead of 0.66.

### 3.2 Antilles heterogeneity

To illustrate the capability to consistently assess a basket of adaption options for different territories with common challenges, the same adaptation measures definition of 3.1.1 can be applied for the neighbouring islands using the indicators of Table 1. The mangrove protection is artificially set by linearly interpolating from the relation of mangroves' area to the islands total area. A maximum of 1.5% and 3% reduction in intensity is fixed on the coast and inland respectively. The first parameter in brackets is the reduction on the coast and the second for inland.

| Island group | Economic growth | Urban growth | Population growth | Mangroves area (ha) | Total area (ha) | Mangrove Protection (%) |
|---|---|---|---|---|---|---|



| | | | | | | |
|---|---|---|---|---|---|---|
| Anguilla | 2.0 | 0.90 | 1.92 | 90 | 9,100 | (0.74, 1.48) |
| Antigua And Barbuda | 2.7 | 0.55 | 1.20 | 700 | 44,000 | (1.19, 2.34) |
| British Virgin Islands | 2.0 | 2.42 | 2.20 | 570 | 15,300 | (1.50, 3.00) |
| Saba And St. Eustatius | 2.0 | 1.00 | 1.00 | 0 | 3,400 | (0, 0) |
| St Barthelemy | 2.3 | 1.00 | 1.00 | 2 | 2,500 | (0.06, 0.12) |
| St Kitts And Nevis | 3.0 | 0.92 | 0.70 | 70 | 26,100 | (0.20, 0.40) |
| St Maarten | 2.1 | 1.56 | 1.39 | 0 | 3,700 | (0, 0) |
| St Martin | 2.3 | 1.00 | 1.00 | 25 | 5,300 | (0.35, 0.71) |
| Turks And Caicos Islands | 3.0 | 1.77 | 2.09 | 23,600 | 61,600 | (1.50, 3.00) |
| US Virgin Islands | 2.0 | 0.10 | 0.00 | 150 | 34,600 | (0.33, 0.65) |

**Table 1: Economic and environmental indicators used in the cost and benefit analysis of adaptation measures per island group. Mangrove protection refers to the percentage of intensity reduced on the cost and in the inland, respectively. The mangrove area (in hectares, ha) is extracted from Food and Agriculture Organization of the United Nations (2007) and the indicators from Central Intelligence Agency (2019).**

With this simple setting the different situations of the islands become apparent. Taking the three most cost-effective measures for each island group and combining them together with the *risk_transfer* option, between 53% and 76% of the total accumulated damages can be averted with a benefit/cost ratio ranging from 0.74 to 0.92. The islands which can avert more than 68% of the damages (represented in gold in Figure 6) manage to do it in different ways. The British Virgin Islands appear to see the most cost-effective measures. By preserving their 570 hectares of mangroves, together with implementing

*preparedness, building_code* and *risk_transfer*, they avert 75% of the total expected damages with a benefit/cost ratio of 0.92. Restoring all the mangroves of the Turks and Caicos Islands appears to be far too expensive, with a benefit/cost ratio of 0.09. The grey options *building_code* and *retrofit* are the ones which, together with *preparedness* and *risk_transfer* manage to reduce 70% of the damages with a befit/cost ratio of 0.80 there. Also Saba and St. Eustatius and Sint Maarten need grey solutions, since they have no mangroves to restore, and achieve to avert 73% and 76% of the expected damages with a benefit/cost ratio

of 0.74 and 0.79 respectively. The amount of mangroves in Anguilla is enough to make it avert 65% of the damages when combined with *preparedness, building_code* and *risk_transfer* (purple category in Figure 6). This is not the case of the United States Virgin Islands, Saint Martin, St Kitts and Nevis and St. Barthelemy, where the same measures lead to a lower reduction of the damage, between 53% and 61% (cyan category in Figure 6). Finally, Antigua and Barbuda manage to avert a similar quantity of expected damage as the later islands with the same measures but has an increased benefit/cost ratio of 0.84.





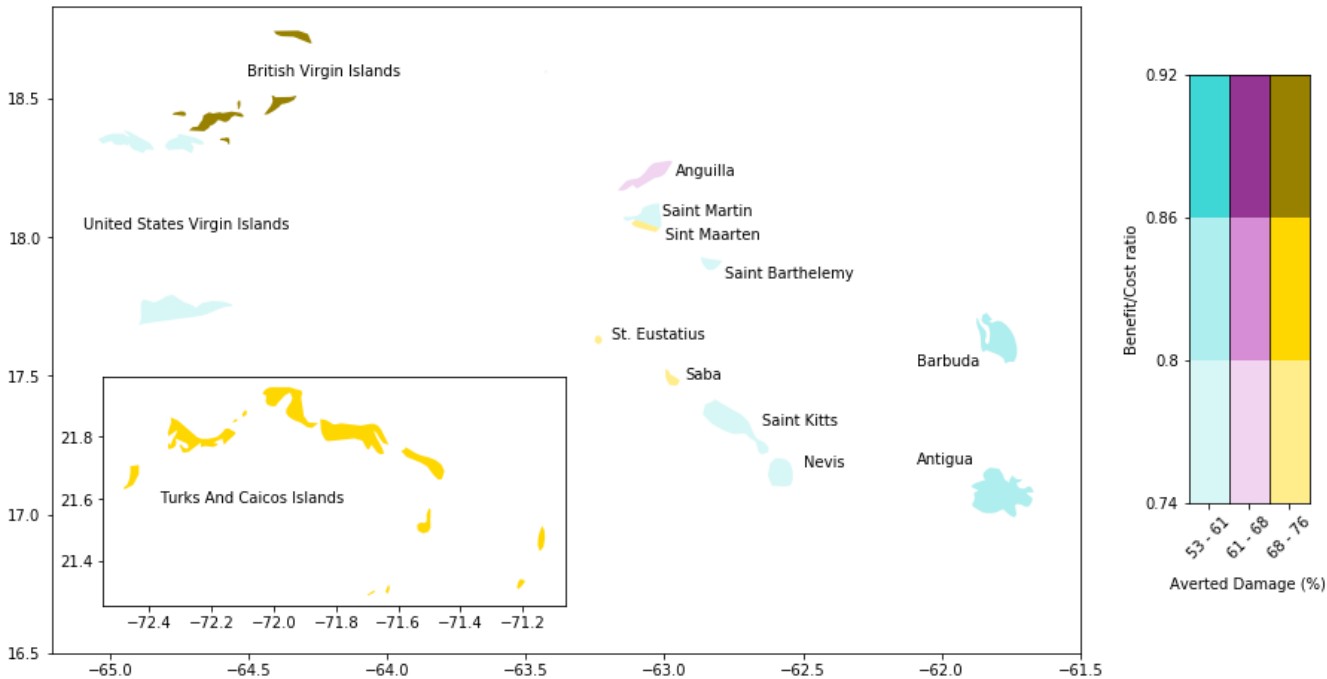

**Figure 6: Cost and benefit relations in Antilles selected islands. The three most cost-effective measures are combined with the risk transfer solution and the resulting net present value of the total expected averted damages from 2016 to 2050 (benefit) is categorized into cyan (53% to 61% damages averted), purple (61% to 68%) and gold (68%-76%). The color intensity represents the benefit/cost ratio: the darkest colors result in more cost-effective measures.**

## 4 Discussion and Outlook

In this paper we presented the concept of probabilistic options appraisal by extension of the impact modelling platform CLIMADA (Aznar-Siguan and Bresch, 2019b). In addition to the application to a specific and bespoke local situation, the platform allows for an intercomparison of adaptation measures across different contexts. CLIMADA underpins the wider Economics of Climate Adaptation (ECA) approach (Souvignet et al., 2016) and offers ready to use global hazard and exposure models able to provide first approximations at local level and consistent regional comparisons. Additionally, high resolution hazard models as well as specific exposure and impact functions can be implemented in CLIMADA to perform detailed analysis on targeted locations and measures, which might have been selected based on the findings of a first less granular analysis. Since CLIMADA integrates an end-to-end view on risk, from the risk drivers such as socio-economic development and climate change scenarios up to the resulting metrics for decision-support, it lends itself to comprehensive sensitivity analyses and allows to identify areas for effective model improvement.

Building on previous work (Aznar-Siguan and Bresch, 2019b) we demonstrate CLIMADA's capabilities to analyse a basket of adaptation options for a set of Caribbean islands hit by hurricane Irma in 2017 by use of openly available indicators. Whilst





an accurate analysis is out of the scope of this paper, the results illustrate the dependence of cost-effective solutions on social and environmental conditions in a limited area and scope of study. While CLIMADA would lend itself to a detailed assessment of uncertainties in all elements of the analysis, we abstained from doing so in order to keep the case study illustrative and the figures more easily readable. We show that combining measures of different nature, such as mangrove restoration,

preparedness, building codes enforcement and retrofitting, can increase the amount of averted damage in a cost-effective way. In Anguilla restoring mangroves averts simulated expected damage of more than 40 m USD over the next 35 years in a moderate climate change scenario, which represents 6% of the scenario's expected damage and more than 4 times the cost estimated for restoration (confirming the finding of a study by the Caribbean Catastrophe Risk Insurance Facility (2010) as well as e.g. Reguero et al. (2018)). Furthermore, combining mangroves restoration together with building codes enforcement

and preparedness increases the averted damage to 14% of the expected damage by 2050 with a cost of approximately one third of the overall benefit. On the other hand, in the Turks and Caicos islands enforcement of building codes results in a more cost-effective measure than mangrove restoration, with a cost ¾ of the benefit. The reason is that, even if both measures avert an expected damage of approximately 220 m USD (11% of the islands expected damage), the restoration of mangroves needs to be implemented just in targeted areas to avert more damage than the invested capital. In these islands the combination of the

grey measures, building codes enforcement and retrofitting, with preparedness education averts 35% of the simulated expected damages in a moderate climate change scenario by 2050 with a high - but effective - cost of 97% of the benefit. In order to avert a significant fraction (more than 50%) of the expected damage over the next 35 years, risk transfer shows to be the most effective complement in all cases studied. Combining insurance – be it indemnity-based or parametric (Caribbean Catastrophe Risk Insurance Facility, 2015) – with other cost-efficient measures reduces its cost (by 50 m USD in the case of Anguilla and

250 m USD in the Turks and Caicos islands) and covers damage of high impact events which cannot be cost-effectively prevented by other measures.

While the idealized case study already provides elements relevant for the development of adaptation strategies and the interplay of prevention, preparedness and risk transfer (c.f. Joyette et al. (2015)), further locally bespoke data would improve the accuracy and representativeness of results, starting from spatially-explicit mapping of specific exposures such as infrastructure

and sectoral split. The concept could handle both indirect impacts (business interruption etc.) as well as series of consecutive events, but we did not venture into these forays for lack of data to validate the model with, similar for other related hazards, where one could expand to torrential rain and separate wind from surge action for tropical cyclones, which would pose a major challenge as there is no separately reported damage and indirect methods would introduce further uncertainties (Strobl, 2012). Methodologically, as mentioned in the introduction, Cost Benefit Analysis (CBA) and Multi-Criteria Analysis (MCA) are not

mutually exclusive, and a comprehensive analysis builds on both used in conjunction with each other. While ECA studies in cities in other regions such as in El Salvador or Bangladesh (Wieneke and Bresch, 2016) did consider asset damage and impacts on people affected and lives lost in their pertinent metrics, some earlier studies also compared health impacts of reduced reservoir outflow (measured in disability adjusted life years as in Finkel (2019)) with hydropower production (measured in MWh and electricity costs) in Tanzania (Bresch and ECA working group, 2009). In order to keep it simple, we did not introduce



metrics other than direct damage in the present case study, with the advantage to easily compare very different adaptation options and their damage aversion potential by reducing all to a common monetary form, but see eminent potential to develop both the platform and its applications in the direction towards MCA, with determination of relative weights of the multiple criteria remaining a challenge. So far, ECA analyses in general and the underlying CLIMADA platform in particular do not

account for a range of critical factors such as the role of institutions, access to and ownership of resources, agency and leadership (Preston and Stafford-Smith, 2009), to name a few. While this will remain a major challenge for long, we do see potential for further development of both the ECA methodology as well as the CLIMADA platform to better serve adaptive management approaches, as they can be used to support the measurement of successful implementation and hence evaluation of proposed adaptation options and therefore provide a framework for lessons learned to inform future actions in an iterative

fashion to make better informed, and often incremental, decisions in the face of uncertainty (Wilby and Dessai, 2010). Such a dynamic simulation platform mitigates shortcomings of static adaptation databases (Mitchell et al., 2016) and lends itself as a basis for web-based adaptation support tools (Glaas et al., 2017) and might hence foster better-informed exchange between the disaster risk management and the climate adaptation (expert) community (Klima and Jerolleman, 2017) not least by informing climate adaptation narratives (Krauß and Bremer, 2020).

**5 Code availability and data availability**

CLIMADA is openly available in GitHub (https://github.com/CLIMADA-project/climada_python, Bresch and Aznar-Siguan (2019a)) under the GNU GPL license (GNU Operating System, 2007). The documentation is hosted in Read the Docs (https://climada-python.readthedocs.io/en/stable/, Aznar-Siguan and Bresch (2019a)) and includes a link to the interactive tutorial of CLIMADA. v1.4.1 was used for this publication, which is permanently available at the ETH Data Archive:

http://doi.org/10.5905/ethz-1007-252 (Bresch et al., 2020). The script reproducing the main results of the paper and all the figures is available under https://github.com/CLIMADA-project/climada_papers (Bresch and Aznar-Siguan, 2019b).

**6 Author contribution**

David N. Bresch conceptualized CLIMADA and oversaw its implementation in Python, based on the previous MATLAB implementation by himself. Gabriela Aznar Siguan designed and executed the Python implementation of the software and did

most of the exemplary case study work.



## 7 Acknowledgements

David N. Bresch developed the first version of CLIMADA as a basis for teaching a master course on uncertainty and risk at ETH back in 2010. He is thankful for many students' feedback and improvement suggestions, without which the model would not be as comprehensive and stable a tool as it is by now.

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
