# Peer review of "CLIMADA v1.4.1: Towards a globally consistent adaptation options appraisal tool"

_Geoscientific Model Development, 2020_

## Referee Comment (RC1) · Anonymous Referee #1 · 28 Aug 2020

The authors present an amendment and application of their open-source python tool CLIMADA to study the benefit of various adaptation options to climate extremes. Their current application deals with a very specific case of tropical cyclone impacts to small islands in the Caribbean. The amendment of the CLIMADA tool seems very useful and timely as it allows to address the full modeling chain from climate impacts to adaptation within a single tool. The code availability and reproducibility on github is best practice. The paper is well motivated and well written despite several very long and complex sentences. Please see my further comments below:

Major points:

1. As mentioned above, the manuscript contains very long and complex sentences that make it difficult for the reader to follow (just to name a few: page 1, line 19; page 14, line 22-28; page 15, line 10). Throughout the text you find many parentheses providing additional information that disturb the flow of reading. Consider to split these long sentences and to provide extra information in additional sentences.

2. Section 2.2.1: This section is very technical and hard to grasp for the non-expert. While I appreciate the discussion along the lines of the actual methods provided in CLIMADA, the reader might get lost easily. It would be helpful to produce a visualization similar to Fig 1 but less technical that summarizes and describes the different methods and their interrelationship. Maybe even a table might be sufficient.

3. Section 3.1.1: It is understood that this section can only provide a rough introduction to the different adaptation measures. However, I think that one needs to be more rigorous and/or comprehensive in order to highlight that CLIMADA is not just a toy model. My comment about uncertainty assessment below points into the same direction. Here are some points one should elaborate on: 1) the impact intensity reduction by mangroves is considered to be 0.74%. This number is given without reference and should be explained. It appears later in Table 1 and seems to be related to the Turks and Caicos Islands, but this remains very opaque. 2) preparedness is set to avoid damages for events with return periods of up to 7 years. Is there some deeper reasoning behind that? Can the authors provide a reference? 3) the paragraph about risk transfer throws around many numbers which are not very well motivated. For instance, it remains unclear to me whether the cost of insurance refers to annual costs or the costs over the whole period.

4. Figure 3 and Figure 4: The reader expects to read off the averted damage (black arrow in Fig. 3) from Fig. 4b. But instead the gap of roughly 300m USD in Fig. 3 corresponds to less than 100m USD in Fig. 4b. Somewhere towards the end of the manuscript it becomes clearer what might have happened: retrofit was neglected. This is rather unsatisfactory, in particular, because the authors claim that combined measures behave differently than single measures. Thus, the reader is unable to reproduce the numbers from the information provided.

5. Section 3.1.3: CLIMADA's ability to combine measures is highlighted in the beginning. When reaching Section 3.1.3 the reader is slightly disappointed as no information about the methods behind the combination is provided (e.g., how is double-counting avoided?). Instead, the reader is confronted with many numbers that require further explanation. In order to better understand how the different measures interact and the numbers come about, I would like to see additional supporting figures in the supplement. Those figures should reproduce the combination effect for the various combinations covered in Section 3.1.3. The figures produced in the jupyter notebook (https://github.com/CLIMADA-project/climada_papers/blob/master/202008_climada_adaptation/reproduce_results.ipynb) should suffice.

6. Section 3.2: What is the reasoning behind choosing the three most cost-effective measures plus risk_transfer? In terms of benefit-cost, this seems not to be the optimal choice based on Fig 4b. Similar to major comment 5, I would also like to see additional supporting figures for all the island groups considered in Figure 6 as a supplement. As above, the figures produced in the jupyter notebook should suffice.

7. Uncertainty assessment: While I understand that uncertainty assessments in this context are very demanding, I still think that the authors need to comment on uncertainties nonetheless. First, in order to strengthen the real-life applicability of CLIMADA, and second, to put the presented numbers into context. The authors cannot extensively discuss benefit-cost ratios with two decimal digits and rate them by effectiveness (fig 6), while claiming in the same instance that uncertainty assessments would overload this paper. I do not want to see an in-depth assessment (knowing the difficulties) but I expect a discussion of the potential sources and ranges of uncertainties for the different measures and how these could affect the presented benefit-cost ratios. This would tremendously help the reader and user to judge on the findings presented in this

manuscript and the possibilities to account for uncertainties using CLIMADA.

Minor points:

1. Abstract: I would find it very useful to mention tropical cyclones as the object of study in the abstract. It remains unclear otherwise against what the discussed adaptation measures for the Caribbean are guarding.

2. Page 1, line 14: basked -> basket

3. Page 3, line 4: the reference "Aznar-Siguan and Bresch 2019" does not appear in the list of references. Please also correct the multiple occurrences of this reference.

4. Page 3, line 10: The (net present value of) the difference ... -> The (net present value of the) difference

5. Page 5, line 6: the concept of mean damage degree (MDD) is mentioned here and throughout the following pages without being defined properly. As MDD is a central concept of this manuscript, I would strongly suggest to explain it on first use. In addition: What is the difference between MDD and mean damage ratio (see Fig. 2)?

6. Page 5, line 28: on -> one

7. Page 5, line 29 (end of line): as well as -> as

8. Page 7, line 6: 2.2.10 -> 2.2.1 ?

9. Page 9, line 20: the reference to Fig 1 seems not correct.

10. Page 9, line 25: the sentence starting with "building code" sounds strange. Are you sure that the 1 m USD mentioned here is correct?

11. Page 10, line 9: Preparedness ... -> By construction, preparedness... I would add this in order to re-iterate that this threshold was chosen at will earlier.

12. Figure 5: The figure is difficult to understand in its current state. 1) I would likely replace the black bars by thicker/colored bars and refer to them as boxes instead of

bars, 2) Reducing alpha for the 40y return period in order to highlight different y scales makes blue the predominant color and confuses the reader. Why don't the authors simply use a vertical line between the 10y and 40y case to highlight the difference between the two bars?

13. Page 11, line 18/19: The mangrove protection discussion is too succinct. Where do the 1.5% and 3% values come from? Why do the Turks and Caicos Islands define the reference? A how is this related to what the reader already knows from section 3.1.1? See also major comment 3.

14. Page 11, line 20: I would transfer the last sentence before the table to the table caption.

15. Cost-benefit vs benefit-cost: Both terms are used interchangeably throughout the manuscript. Please stick to one wording and adjust also Fig 6 accordingly.

16. Figure 6: Which scenario is displayed here? Current or moderate future?

---

## Referee Comment (RC2) · Anonymous Referee #2 · 18 Sep 2020

Gabriela Aznar-Siguan

**Anonymous Referee #2**

The authors intend to introduce a methodology that integrate climate modelled risk, impacts (loss and damage), and adaptation options assessment (cost/benefit analysis). In addition, they provide a case study in Antilles to demonstrate an example to use the tool. The intentions are valuable and the platform seems useful to scientists and decision makers at local levels. However, the authors fail to present their intentions and execution well enough for readers to comprehend the value of this study. Here are my comments to this paper:

1.The paper is difficult to read because of a lot of grammar issues. It is perhaps better to proofread the entire text in the next revision.

[Figure]

2.Section 1 (Introduction): This section is mixed with problem statement and literature review, which make this section confusing. Unfortunately, both (problem statement and theoretical background) are not presented clearly. What's the problem now? What's the scientific gap now? What does this study aim to achieve? These questions can help readers to get to know the reasons behind this study. In addition, a lot of reviewed literature are citing the authors' previous work and stating the content of the reviewed papers. It lacks of discussion of the problems of current practices from reviewing literature.

3.Section 2 (Framework Concept and Design): This section provides a lot of technical details of CLIMADA. It is useful to add some important perspectives. For example, can CLIMADA be used in every climate impacts? The paper uses Hurricane as an example risk, but can other impacts (e.g., agriculture, health, etc.) be used in the platform? Is there a constrain in this tool? Such as data availability? In addition, why a moderate scenario is selected? Since the authors are exploring a hazard/disaster impact, why not use the worst case scenario (RCP 8.5)?

4.Section 3 (Case Study): It is perhaps helpful if the authors can provide some background information of current response measures of Antilles in facing Hurricane hazards. In addition, one key challenge of climate modeling in island nation is the resolution and hurricane projection. Did you conduct downscaling? How did you project hurricanes in 2050?

---

## Author Comment (AC1) · 31 Oct 2020

**AUTHORS' RESPONSE TO REFEREE #1**

**Research article:**

Bresch, D. N. and Aznar-Siguan, G.: CLIMADA v1.4.1: Towards a globally consistent adaptation options appraisal tool, Geosci. Model Dev. Discuss., https://doi.org/10.5194/gmd-2020-151

**Authors:**

David N. Bresch (dbresch@ethz.ch), Gabriela Aznar-Siguan (Gabriela.Aznar@meteoswiss.ch)

*We thank the anonymous referee for his comments, which have improved the quality of the manuscript.*

*The original comments from the referee are listed below directly followed by our responses in blue and italic and changes to the manuscript in blue and **bold** (unless where it gets complicated or tiny, where changes are made in the manuscript only).*
* * *
The authors present an amendment and application of their open-source python tool CLIMADA to study the benefit of various adaptation options to climate extremes. Their current application deals with a very specific case of tropical cyclone impacts to small islands in the Caribbean. The amendment of the CLIMADA tool seems very useful and timely as it allows to address the full modeling chain from climate impacts to adaptation within a single tool. The code availability and reproducibility on github is best practice. The paper is well motivated and well written despite several very long and complex sentences. Please see my further comments below:

Major points:

1. As mentioned above, the manuscript contains very long and complex sentences that make it difficult for the reader to follow (just to name a few: page 1, line 19; page 14, line 22-28; page 15, line 10). Throughout the text you find many parentheses providing additional information that disturb the flow of reading. Consider to split these long sentences and to provide extra
information in additional sentences.

*We took care of splitting sentences where appropriate (quite some instances, indeed) and did*
*move (longer) remarks in brackets into full sentences to increase readability as suggested.*
*Please find all in the track change version of the revised manuscript rather than listing all*
*changes here.*

2. Section 2.2.1: This section is very technical and hard to grasp for the non-expert. While I
appreciate the discussion along the lines of the actual methods provided in CLIMADA, the
reader might get lost easily. It would be helpful to produce a visualization similar to Fig 1 but
less technical that summarizes and describes the different methods and their interrelationship.
Maybe even a table might be sufficient.

*The UML diagram displayed in Fig 1 helps to locate the main classes and to understand their*
*relation. It extends Fig 1 of the previous paper Aznar-Siguan & Bresch 2019b. We decide*
*therefore to modify this figure instead of inserting a new one.*

*We have modified Fig 1 to include all the methods and attributes described in Section 2.2.1 and*
*2.2.2, since many of them were not represented before. These are:*

• *In CostBenefit class the methods combine_measures, apply_risk_transfer,*
*plot_cost_benefit and plot_event_view*
• *In Measures class the attributes cost, exp_region_id, hazard_set, hazard_freq_cutoff,*
*exposures_set, imp_fun_map, mdd_impact, paa_impact, risk_transf_cost_factor,*
*risk_transf_attach and risk_transf_cover.*

3. Section 3.1.1: It is understood that this section can only provide a rough introduction to the
different adaptation measures. However, I think that one needs to be more rigorous and/or
comprehensive in order to highlight that CLIMADA is not just a toy model. My comment about
uncertainty assessment below points into the same direction. Here are some points one should
elaborate on: 1) the impact intensity reduction by mangroves is considered to be 0.74%. This
number is given without reference and should be explained. It appears later in Table 1 and seems
to be related to the Turks and Caicos Islands, but this remains very opaque. 2) preparedness is set
to avoid damages for events with return periods of up to 7 years. Is there some deeper reasoning
behind that? Can the authors provide a reference? 3) the paragraph about risk transfer throws around many numbers which are not very well motivated. For instance, it remains unclear to me whether the cost of insurance refers to annual costs or the costs over the whole period.

*How realistic the adaptation measures are in CLIMADA depends only on the input data and/or*

*models of each specific case. CLIMADA does not provide "default" measures but several ways*

*of parametrizing and comparing them. The parametrizations presented here have been chosen to*

*reproduce the main findings on Anguilla in Caribbean Catastrophe Risk Insurance Facility*

*(2010), where CLIMADA was used together with field data. This analysis uses only openly*

*available data and, as such, can be used as a preliminary study to select which measures could*

*be considered and further modelled with local data. This is stressed in Section 4:* "While the idealized case study already provides elements relevant for the development of adaptation strategies and the interplay of prevention, preparedness and risk transfer (c.f. Joyette et al.

(2015)), further locally bespoke data would improve the accuracy and representativeness of results, starting from spatially-explicit mapping of specific exposures such as infrastructure and sectoral split."

*We add a sentence in Section 3.1.1:* "The parametrizations chosen here allow to reproduce the main findings on Anguilla in Caribbean Catastrophe Risk Insurance Facility (2010)". We modify as well the abstract to clarify the scope of this case study as follows: "We apply the open-source

Python implementation to a tropical cyclone impact case study in the Caribbean **with openly**

**available data**. This allows to prioritize a small basket of adaptation options, namely green and grey infrastructure options as well as behavioural measures **and risk transfer**, and permits inter- island comparisons. In Anguilla, for example, mangroves avert simulated damages more than 4

times the cost estimated for restoration, while enforcement of building codes shows to be effective in the Turks and Caicos islands **in a moderate climate change scenario**."

*Ad 1) Factor 0.74 provides sensible results for Anguilla (based on Caribbean Catastrophe Risk*

*Insurance Facility, 2010) and is used as reference to interpolate linearly the factors of the other*

*islands according to their ratio of mangrove area to island area.*

*We have rephrased the explanation in Section 3.2:* "The mangrove protection is set by linearly interpolating Anguilla's factor proportionally to the island's ratio of mangroves' area to total area."

*Ad 2) This criteria is set to show the effect of such a threshold on the computations. It is set to avoid damages of events generating less than 1.5 m USD.*

*We have rephrased the explanation in Section 3.2:* "This measure ideally reduces the effective wind intensity and avoids most of the damages for events with low return periods. We set a wind intensity reduction of 0.5% (see impact function in Figure 2) and a threshold of 7 year events under which no damages are generated. This threshold corresponds to events with exceedance damages lower than 1.5 m USD."

*Ad 3) Again, to the illustrative character of the study, the exact definition of the risk transfer "layer" (as it is called in the insurance industry) is inspired by business practice, yet other levels of attachment (instead of 12 years) and cover (instead of up to a return period of 145 years) are equally well possible. The numbers chosen are realistic for a multi-island scheme (such as CCRIF). Please note that the Jupyter notebook (see points 5 and 6 below) does allow to experiment with these settings.*

*We hence clarified as follows:* "Finally, risk transfer is considered, being particularly suitable to manage risks of low frequency, high severity events. We define an insurance layer with attachment point (or deductible, **i.e. the damage amount corresponding to a frequency at which the risk transfer gets triggered on average**) and cover (**the amount of damage covered by risk transfer**) proportional to the island's expected exceedance damages. The attachment is set to the 12 year per event damage (approximately 32 m USD) and the cover **designed such as to cater for events with up to a** 145 year return period (**the risk transfer thus covering** approximately 314 m USD **per event**)."

*As for costs of measures, all are net present value (NPV) over the whole period, in order to compare also with NPV of averted damage over the whole period. We clarified in the manuscript by adding on page 5 after the explication about risk transfer costs:* **"For risk transfer therefore, the cost is calculated by CLIMADA."**.

4. Figure 3 and Figure 4: The reader expects to read off the averted damage (black arrow in Fig. 3) from Fig. 4b. But instead the gap of roughly 300m USD in Fig. 3 corresponds to less than 100m USD in Fig. 4b. Somewhere towards the end of the manuscript it becomes clearer what might have happened: retrofit was neglected. This is rather unsatisfactory, in particular, because the authors claim that combined measures behave differently than single measures. Thus, the
reader is unable to reproduce the numbers from the information provided.

*The combination of a subset of measures is represented in Figure 3 to represent the fact that*
*only a selection of the studied measures are eventually implemented due to budget limitations*
*and other constraints (see also point 6 below).*

*We add a label with the name of the measures represented in Figure 3 and modify its caption*
*using "**combining** the measures" instead of "**implementing** the measures".*

5. Section 3.1.3: CLIMADA's ability to combine measures is highlighted in the beginning.
When reaching Section 3.1.3 the reader is slightly disappointed as no information about the
methods behind the combination is provided (e.g., how is double-counting avoided?). Instead,
the reader is confronted with many numbers that require further explanation. In order to better
understand how the different measures interact and the numbers come about, I would like to see
additional supporting figures in the supplement. Those figures should reproduce the combination
effect for the various combinations covered in Section 3.1.3. The figures produced in the jupyter
notebook
([https://github.com/CLIMADAproject/climada_papers/blob/master/202008_climada_adaptation/](https://github.com/CLIMADAproject/climada_papers/blob/master/202008_climada_adaptation/)
[reproduce_results.ipynb](reproduce_results.ipynb)) should suffice.

*Thanks for pointing to this issue. Combine measures primarily means that double-counting of the*
*simple kind is avoided. In combining benefits, the combined benefit can never amount to more*
*than the damage itself. As CLIMADA is fully event based, the benefit of each measure is first*
*calculated independently for each event. In a second step, the benefits of say two measures are*
*added for each event and it is ensured that this sum never exceeds the damage without measures*
*(i.e. combined measures can maximally avoid any damage). Combinations of the second kind,*
*i.e. synergies or dis-synergies are not modelled. A synergy would mean two measures lead to*
*higher a benefit than the sum of benefits, as could be the case when combining e.g. an early*
*warning system with an evacuation plan (e.g. in the Bangladesh case study, Wieneke and Bresch,*
*2016). Dis-synergies lead to a reduction of the combined benefit (see the following illustration).*

[Figure]

*Illustration of dis-synergies (unpublished backup material of the Samoa case study, as*

*summarized in Bresch, D. N. and ECA working group, 2009,*

*https://ethz.ch/content/dam/ethz/special-interest/usys/ied/wcr-*

*dam/documents/Economics_of_Climate_Adaptation_ECA.pdf#page=110). A detailed treatment*

*of this complex interplay of measures is far beyond the scope of the present paper, albeit we did*

*use a – precursor of – CLIMADA for this figure.*

*Instead of adding a supplement or appendix, we decided to make available the full Jupyter*

*notebook, which allows for a reproduction of the detailed results, an inspection of specific*

*parameters and – if CLIMADA is locally installed, even for an interactive change of parameters*

*and settings. We therefore added the reference to the Jupyter notebook in the references as:*

*Aznar-Siguan, G. and Bresch, D. N.: CLIMADA - Caribbean case study. Jupyter notebook, 2020.*

*https://github.com/CLIMADA-*

*project/climada_papers/blob/master/202008_climada_adaptation/reproduce_results.ipynb [last*

*retrieved 24 Oct 2020] and added the reference in page 11 as:* "**Please find the detailed results**

**in Aznar-Siguan and Bresch, 2020.**"

*Please note further that we present a case study and hence numbers are illustrative. Therefore, a*

*lengthy appendix could far less serve the purpose of providing exemplary insights compared to*

*the notebooks – and numbers from the appendix would not be of much use in isolation either.*

6. Section 3.2: What is the reasoning behind choosing the three most cost-effective measures plus risk_transfer? In terms of benefit-cost, this seems not to be the optimal choice based on Fig

4b. Similar to major comment 5, I would also like to see additional supporting figures for all the island groups considered in Figure 6 as a supplement. As above, the figures produced in the jupyter notebook should suffice.

*We chose a set of three measures merely for illustrative purposes. With the combination of*

*measures being treated in an approximate way, we would like to show how far one can get by*

*using this features, especially to explore the effectiveness of risk transfer. Risk transfer costs are*

*substantially lowered by any (combination of) adaptation measures. We decided not to combine*

*all four measures to mimic the budgetary constraints one might encounter in a real case. It needs*

*to be noted further that in many of the real case studies the authors have been involved in, sets of*

*measures were often built more on a multi-criteria (MCA) rather than a purely CBA approach.*

*But given the purely illustrative purpose of the present case study, we do not venture into this*

*here.*

*We chose risk transfer to exemplify the risk-reducing benefit of measures translating into*

*considerable reduction in risk transfer costs, a point (very) relevant to the Caribbean Cat Risk*

*Insurance Facility (CCRIF, 2010) and its offering to strengthen societal resilience in the region.*

*But we did abstain from modelling the proper scheme (index based etc.) again for the sake of*

*simplicity of the case study provided. As a side remark, we are currently working on a study*

*applying CLIMADA to the cash-out structure of the European Stability Fund (ESF) in the*

*Caribbean region (as there are European liabilities) ...*

*As for the details about combining measures, the Jupyter notebook does provide the detailed*

*results for all islands and we deem it (as in point 5) more suitable to provide direct access to the*

*notebook (on GitHub, maintained, even versioned) rather than adding lengthy tables in an*

*appendix. We therefore added to the text at the bottom of page 12 as follows:* **"Detailed results**

**per island as well as the possibility to further experiment with different parameters/settings**

**can be found in Aznar-Siguan and Bresch, 2020."**

7. Uncertainty assessment: While I understand that uncertainty assessments in this context are very demanding, I still think that the authors need to comment on uncertainties nonetheless.

First, in order to strengthen the real-life applicability of CLIMADA, and second, to put the presented numbers into context. The authors cannot extensively discuss benefit-cost ratios with two decimal digits and rate them by effectiveness (fig 6), while claiming in the same instance that uncertainty assessments would overload this paper. I do not want to see an in-depth assessment (knowing the difficulties) but I expect a discussion of the potential sources and ranges of uncertainties for the different measures and how these could affect the presented benefit-cost ratios. This would tremendously help the reader and user to judge on the findings presented in this manuscript and the possibilities to account for uncertainties using CLIMADA.

*We truly appreciate your kind understanding that a comprehensive uncertainty assessment in*

*this context would be very demanding. We are currently exploring non-standard (beyond brute-*

*force Monte-Carlo approaches), but these early stage experiments with CLIMADA do in fact*

*exceed the scope of the present paper.*

*We agree that it does not make sense to state unnecessary mock precision in benefit/cost ratios.*

*Indeed, in the right part of Figure 6, we aim at illustrating the fact that islands can be grouped*

*and the second digit merely stems from labelling the vertical axis. We had a version with*

*rounded figures, but felt it looked awkward. To clarify, we amended to the legend of Figure 6 as*

*follows*: "The three most cost-effective measures are combined with the risk transfer solution and the resulting net present value of the total expected averted damages from 2016 to 2050 (benefit)

is categorized into **three equally spaced ranges**, cyan (53% to 61% damages averted), purple (61% to 68%) and gold (68%-76%) **and Benefit/Cost ratio is also shown in three indicative**

**ranges**. The color intensity represents the benefit/cost ratio: the darkest colors result in more cost-effective measures."

*As for a discussion of the potential sources and ranges of uncertainties for the different measures*

*and how these could affect the presented benefit-cost ratios, we added the following in the*

*Discussion:*

**"Main drivers of uncertainty, beyond those in hazard, exposure, and vulnerability (Aznar-**

**Siguan and Bresch, 2019b) for the four adaptation measures, while not quantified, can at**

**least be qualitatively described as follows. As for preparedness, the level and scope for this**

**study have been chosen based on general findings of previous ECA studies (Caribbean Cat**

**Risk Insurance Facility, 2010), where large differences had been found across regions,**

**mainly stemming from barriers to implementation, not least such as lack of agency of non-**

**owner property residents. Notwithstanding, in all cases, preparedness does lower damages**

**and almost always at a Benefit/Cost ratio >1 on a societal level, which does not necessarily**

mean it being 'worth the money' for the single property owner each time. As for mangroves, differences of applicability to single islands have been mentioned above. Again, as shown in studies (Reguero et al., 2018), such nature-based solutions, while difficult to assess at great precision in terms of exact Benefit/Cost yield ratios far above one if applied at scale. With building codes, it all depends on design - and enforcement. The latter being utterly cultural, any assessment must remain spurious, even past experience might not provide solid a guidance for present and future uptake. On the other hand, implementation is rather straightforward in CLIMADA in terms of the impact function as far as the design component is concerned, hence relative uncertainty can be limited there. Retrofit is implemented the exact same way as building codes and exposed to the same threat of enforcement. Risk transfer in contrast to measures discussed so far, being a purely monetary transaction, does, in its assessment at least, suffer from far less uncertainty. But it inherits all the underlying uncertainty of the probabilistic model as well as of the measures in terms of their risk-reducing capacity. Testing with many (sets of) parameters (Aznar-Siguan and Bresch, 2020), results regarding the effectiveness of risk transfer proved robust."

Minor points:

1. Abstract: I would find it very useful to mention tropical cyclones as the object of study in the abstract. It remains unclear otherwise against what the discussed adaptation measures for the Caribbean are guarding.

*This point is very valid, thanks for bringing this to our attention. We added to the Abstract as follows:* "We apply the open-source methodology and its Python implementation to a **tropical cyclone impact** case study in the Caribbean, which allows to prioritize a small basket of adaptation options, [...]"

2. Page 1, line 14: basked -> basket

*Corrected*

3. Page 3, line 4: the reference "Aznar-Siguan and Bresch 2019" does not appear in the list of references. Please also correct the multiple occurrences of this reference.

*Thanks. That's in fact "Aznar-Siguan and Bresch 2019**b**", corrected.*

4. Page 3, line 10: The (net present value of) the difference : : : -> The (net present value of the)
difference

*Corrected*

5. Page 5, line 6: the concept of mean damage degree (MDD) is mentioned here and throughout
the following pages without being defined properly. As MDD is a central concept of this
manuscript, I would strongly suggest to explain it on first use. In addition: What is the difference
between MDD and mean damage ratio (see Fig. 2)?

*Aznar-Siguan and Bresch 2019b describe this in detail, hence we clarified as follows:*

"Even if new impact functions can be easily introduced, the following attributes allow to perform
linear transformations to given impact functions: hazard_inten_imp transforms the abscissae
(e.g. implementing elevation of homes in the case of flood) while  mdd_impact and paa_impact
transform, respectively, the Mean Damage Degree **(MDD)** and the Percentage of Affected Assets
**(PAA,** e.g. to reflect an improved building code). **Please note that the Mean Damage Ratio**
**(MDR) is defined as the product of MDD and PAA for any given intensity, see Aznar-**
**Siguan and Bresch (2019b) for a detailed description.**"

6. Page 5, line 28: on -> one

*Corrected*

7. Page 5, line 29 (end of line): as well as -> as

*We replaced* "as well as" *by* "**and**", *which makes it more lisible.*

8. Page 7, line 6: 2.2.10 -> 2.2.1 ?

*That's strange, as it reads 2.2.1 in the Word file, but got wrongly stated in the pdf generated. We*
*now checked again in the revised pdf and resolved this.*

9. Page 9, line 20: the reference to Fig 1 seems not correct.

*Indeed it should refer to Figure 3, corrected.*

10. Page 9, line 25: the sentence starting with "building code" sounds strange. Are you sure that the 1 m USD mentioned here is correct?

*Thanks for pointing this out, we clarified as follows:*

"Building_code averts **order of** 1 m USD **of damage more** than mangrove, but its benefit/cost ratio stays bellow 2."

11. Page 10, line 9: Preparedness : : : -> By construction, preparedness: : : I would add this in order to re-iterate that this threshold was chosen at will earlier.

*Good point, adjusted as* "**By construction***, preparedness* averts …"

12. Figure 5: The figure is difficult to understand in its current state. 1) I would likely replace the black bars by thicker/colored bars and refer to them as boxes instead of bars, 2) Reducing alpha for the 40y return period in order to highlight different y scales makes blue the predominant color and confuses the reader. Why don't the authors simply use a vertical line between the 10y and 40y case to highlight the difference between the two bars?

*Ad 1) The plot shows bars (as a bar chart) and colored boxes. The black bars represent the total*

*exceedance damage for events with the corresponding return period. The colored boxes*

*represent the amount of damage that can be averted by the corresponding measure. The last*

*ones are "boxes" or "blocks" in the sense that their height and not their y-value is the averted*

*damage.*

*Ad 2) Agree*

*We remove alpha and add the suggested line in Figure 5. Further, we clarified the figure caption*

*as follows:* "Averted impact of each measure in 2050 for different return periods, without taking into account climate change nor economic growth (a) and with the moderate risk increase (b).

The **thin** black bars show the expected exceedance damage at each return period and each coloured block indicates the amount averted by the corresponding measure. **The capacity of**

**measures to absorb damage exceeds expected damage for high frequency (7 year) events**

**and risk transfer is more than sufficient in a)**. Note …"

13. Page 11, line 18/19: The mangrove protection discussion is too succinct. Where do the 1.5%

and 3% values come from? Why do the Turks and Caicos Islands define the reference? A how is this related to what the reader already knows from section 3.1.1? See also major comment 3.

*See our response to point 3 above, where we take this in account, too.*

14. Page 11, line 20: I would transfer the last sentence before the table to the table

*This formatting suggestion is well taken, we took care of.*

---

## Author Comment (AC2) · 31 Oct 2020

**AUTHORS' RESPONSE TO REFEREE #2**

**Research article:**

Bresch, D. N. and Aznar-Siguan, G.: CLIMADA v1.4.1: Towards a globally consistent adaptation options appraisal tool, Geosci. Model Dev. Discuss., https://doi.org/10.5194/gmd-2020-151

**Authors:**

David N. Bresch (dbresch@ethz.ch), Gabriela Aznar-Siguan (Gabriela.Aznar@meteoswiss.ch)

*We thank the anonymous referee for his comments, which have improved the quality of the manuscript.*

*The* original comments from the referee are listed below *directly followed by our responses in blue and italic and changes to the manuscript in blue and* **bold** *(unless where it gets complicated or tiny, where changes are made in the manuscript only).*
* * *
September 2020

The authors intend to introduce a methodology that integrate climate modelled risk, impacts (loss and damage), and adaptation options assessment (cost/benefit analysis). In addition, they provide a case study in Antilles to demonstrate an example to use the tool. The intentions are valuable and the platform seems useful to scientists and decision makers at local levels. However, the authors fail to present their intentions and execution well enough for readers to comprehend the value of this study. Here are my comments to this paper:

1.The paper is difficult to read because of a lot of grammar issues. It is perhaps better to proofread the entire text in the next revision.

*We carefully re-checked the paper and adopted a more lisible style throughout – in line also with the other reviewer's remark about occasionally long sentences (sic). With the many changes to the text, we do not list all of them here, but provide both a clean revised version of the paper as well as a version with track changes.*

2.Section 1 (Introduction): This section is mixed with problem statement and literature review,
which make this section confusing. Unfortunately, both (problem statement and theoretical
background) are not presented clearly. What's the problem now? What's the scientific gap now?
What does this study aim to achieve? These questions can help readers to get to know the
reasons behind this study. In addition, a lot of reviewed literature are citing the authors' previous
work and stating the content of the reviewed papers. It lacks of discussion of the problems of
current practices from reviewing literature.

*The introduction of the paper is structured along the following 'fil rouge': Climate change is a*
*fact, yet greenhouse gas mitigation does not happen at the required scale, hence the need for*
*adaptation and demand for risk assessment and adaptation options appraisal. Adaptation is*
*(utterly) local, but best informed by globally consistent approaches.*

*In line with point 1, we broke many sentences in two, reformulated as appropriate and better*
*highlighted the 'fil rouge' also by breaking the introduction into sections. Again, as for point 1,*
*given the many changes, we do not list all of them here, but provide both a clean revised version*
*of the paper as well as a version with track changes.*

*The gap consists in the mere fact that globally consistent approaches to adaptation options*
*appraisal are rare to non-existent – and none are readily available as an open-source and -*
*access software tool.*

*Hence the need to set globally consistent approaches forth, underpinned by versatile platforms,*
*ready for practical application. In this sense, the introduction states the clear demand and does*
*not focus on an in-depth discussion of current practices, as this is not the aim of the present*
*paper. We deemed it useful to cite key contributions to support our argumentation, but do not*
*aim at a review of the full body of literature, which would warrant a study of its own.*

*The aim of the paper is to present the open-source and -access CLIMADA platform which*
*implements the Economics of Climate Adaptation (ECA) framework, as described in the last*
*paragraph of the introduction. Hence we deem it useful to provide the basics about ECA, which*
*leads to citing a couple of previous studies.*

*We carefully reviewed and removed select references as suggested. As we strived to keep the*
*paper to the point, we provide a brief description of ECA in the introduction, too, such that we*
*can focus in CLIMADA in the methods section of the paper.*

*Having thus laid out the structure of the paper, we deem it useful to end the introduction with a*
*sentence to stress the enabling nature of this work. Again, we set this apart by introducing a*
*break to separate from the signposting in the sentence before. Still, we deem it helpful to stress*
*the enabling function of this work already at the end of the introduction, not only to conclude the*
*paper with, namely:*

"This extended version of the CLIMADA platform has been designed to enable risk assessment
and options appraisal in a modular form and occasionally bespoke fashion […] yet with high
reusability of common functionalities to foster usage in interdisciplinary studies […] and
international collaboration."

3.Section 2 (Framework Concept and Design): This section provides a lot of technical details of
CLIMADA. It is useful to add some important perspectives. For example, can CLIMADA be
used in every climate impacts? The paper uses Hurricane as an example risk, but can other
impacts (e.g., agriculture, health, etc.) be used in the platform? Is there a constrain in this tool?
Such as data availability? In addition, why a moderate scenario is selected? Since the authors are
exploring a hazard/disaster impact, why not use the worst case scenario (RCP 8.5)?

*While the present application focuses for purely illustrative purposes on hurricane risk in the*
*Caribbean, the CLIMADA platform* can *not only, but* is *actually used for most extreme weather*
*events in a globally consistent manner. To clarify this point, we therefore added to the*
*manuscript:* "**Please note that CLIMADA does provide global coverage of major hazards**
**beyond tropical cyclones (TC), yet we focus in TC in the present paper for illustrative**
**purposes.**"

*As of today, CLIMADA provides global coverage of all major climate-related extreme-weather*
*hazards at high resolution, namely (i) tropical cyclones and storm surge at 10 and 1km, (ii) river*
*flood at 4km, (iii) drought at 50km, (iv) wildfire at 1km and (v) European winter storms at 4km.*
*Tropical cyclones (Geiger at al., 2019; ) are based on IBTrACS (Knapp et al., 2010; updated*
*monthly since)., river flood (Sauer et al., submitted) and drought (Eberenz et al., in preparation)*
*on isimip (isimip.org), European winter storms on Copernicus WISC (Welker et al., submitted)*

*and wildfires on MODIS (https://modis.gsfc.nasa.gov, implementation experimental still). For all*

*mentioned hazards, a historic, a probabilistic and several future climate (RCP-based) hazard*

*sets exist, enabling assessment of risks today and under diverse climate scenario futures.*

*CLIMADA does also provide a globally consistent exposure dataset at 1km resolution, based on*

*population and satellite-measured night-light intensity (Eberenz et al., 2020a). To implement*

*bespoke vulnerability, impact functions have been calibrated for global regions for tropical*

*cyclones (Eberenz et al., 2020b, in review), flood (Sauer et al., submitted) and European winter*

*storms (Welker et al., submitted). With hazard, exposure and vulnerability datasets being*

*provided, CLIMADA is currently the only ready to use open-source and access (no strings*

*attached, even free for commercial use, GNU GPL license ) globally consistent impact modeling*

*platform.*

*Sure there are constraints, but given the versatility of the general concept as well as the*

*openness of the platform itself, it is merely available extreme weather hazard data that limits its*

*use. While the paper focuses on a regional application, the platform has been used an many*

*scales, from global (e.g. Gettelman et al. 2017) to truly local (c.f. Wieneke and Bresch, 2016).*

*As for the scenario, again, we chose this for illustrative purposes, any other combination of RCP*

*and year, can, based on Knutson et al. 2015, readily be explored. See also last para of the*

*answer to the next point.*

*One can play with the RCP selection (and other parameters/settings) in the Jupyter notebook as*

*provided - we will add this as a reference to the paper, instead of a static appendix*

*(https://github.com/CLIMADAproject/climada_papers/blob/master/202008_climada_adaptation/repro*

*duce_results.ipynb).*

*As we intend to use the case study in many conversations, not all are best initiated with the worst*

*case to start with – hence we would like to trigger questions exactly such as yours (why not*

*RCP8.5) rather than impose this. In this sense, too, your comment is highly appreciated.*

4.Section 3 (Case Study): It is perhaps helpful if the authors can provide some background information of current response measures of Antilles in facing Hurricane hazards. In addition, one key challenge of climate modeling in island nation is the resolution and hurricane projection.

Did you conduct downscaling? How did you project hurricanes in 2050?

*With the case study being illustrative, it was by no means within the scope of the present paper to*
*study the local situation in terms of actually implemented response measures. But we welcome*
*the comment in the spirit of the many Economics of Climate Adaptation (ECA) studies we*
*conducted so far in most world regions, with teams on the ground and deeply rooted in a*
*transdisciplinary approach both in shaping and scoping of the studies. Given limited resources,*
*efforts were directed at contributing facts suitable for local decision making and technical*
*reports, rather than bringing these studies into the peer-reviewed body of literature. This was*
*also due to the fact that at that time, the first author was fully employed in a private sector*
*company with global presence and local attention. Other priorities kept him from publishing in*
*other forms than technical reports (see https://wcr.ethz.ch/research/casestudies.html for a*
*collection) and policy briefs, such as e.g. to the G20 (World Bank Group, 2017).*

*No downscaling was employed in the study, as the probabilistic tropical cyclone track set was*
*modified according to on Knutson et al. 2015. The wind fields, calculated based on Holland*
*(2008) can be calculated at any spatial resolution, down to 1 km is reasonable. Again, as we*
*present an illustrative case for the full options appraisal methodology, any (sub)model can be*
*further refined, the tropical cyclone wind field e.g. by adding a surface roughness component to*
*it, the exposure layer by specifying sector-specific exposure etc.*

*For the climate projection 2050, we applied the Atlantic basin factors as published by Knutson et*
*al. 2015 to the probabilistic tropical cyclone track set, i.e. we modified the single event*
*frequency and wind field intensity accordingly. Specifically, we multiplied the wind intensity of*
*storms with category greater than 1 by a factor of 1.045, interpolating these values between the*
*time stamps, and left the event frequency unchanged, all as provided by Knutson et al. 2015 table*
*3 (we just consider changing frequencies and intensities when the significance level of the*
*hypothesis test is lower than 0.05). Again, we would like tom stress the fact the case study is*
*provided as an illustrative example, by no means pre-empting other methods to generate hazard*
*datasets, such as e.g. obtaining tracks from GCMs (as done in Gettelman et al. 2017) or hybrid*
*methods, such as using synthetic tracks (Geiger et al. 2018), both papers employing CLIMADA*
*for all impact calculations.*

*References as used in this reply (which are not referenced in the paper itself):*

Eberenz, S., Stocker, D., Röösli, T., and Bresch, D. N., 2020a: *Asset exposure data for global physical risk assessment, Earth Syst. Sci. Data*, 12, 817–833, *https://doi.org/10.5194/essd-12-817-2020*.

Eberenz, S., Lüthi, S., and Bresch, D. N., 2020b: *Regional tropical cyclone impact functions for globally consistent risk assessments, Nat. Hazards Earth Syst. Sci. Discuss.*, *https://doi.org/10.5194/nhess-2020-229*. *In review.*

Geiger, T., Frieler, K., and Bresch, D. N., 2018: *A global historical data set of tropical cyclone exposure (TCE-DAT), Earth Syst. Sci. Data*, 10, 185–194, 2018. *https://www.earth-syst-sci-data.net/10/185/2018/essd-10-185-2018.pdf* , DOI: *https://doi.org/10.5194/essd-10-185-2018*

Gettelman, A., Bresch, D. N., Chen, C. C., Truesdale, J. E., Bacmeister, J. T., 2017: *Projections of future tropical cyclone damage with a high-resolution global climate model. Climatic Change*, 146, 3–4, pp 575–585. *https://doi.org/10.1007/s10584-017-1902-7*

Knapp, K. R., Kruk, M. C., Levinson, D. H., Diamond, H. J., and Neumann, C. J., 2010: *The international best track archive for climate stewardship (IBTrACS), B. Am. Meteorol. Soc.*, 91, 363–376, *https://doi.org/10.1175/2009BAMS2755.1*.

Sauer, I., Reese, R., Otto, C., Geiger, T., Willner, S. N., Guillod, B., David N. Bresch and Frieler, K.: *Climate induced trends and variability in observed river flood damages. Submitted.*

Welker, C., Röösli, T., Bresch, D. N.: *Risk assessment of building damages associated with extreme winter windstorms in the canton of Zurich, Switzerland. Submitted.*

Wieneke, F., & Bresch, D. N., 2016: *Economics of Adaptation (ECA) in Development Cooperation: A Climate Risk Assessment Approach Supporting decision making […]. Materials on Development Financing, UNU, KfW. https://www.kfw-entwicklungsbank.de/PDF/Download-Center/Materialien/2016_No5_Economics-of-Adaptation_EN.pdf*

World Bank Group, 2017. *Sovereign Catastrophe Risk Pools: World Bank Technical Contribution to the G20. World Bank, Washington, DC. © World Bank. https://openknowledge.worldbank.org/handle/10986/28311*

---

## Author Response (AR2)

**AUTHORS' RESPONSE TO REFEREE #1, SECOND ROUND**

**Research article:**

Bresch, D. N. and Aznar-Siguan, G.: CLIMADA v1.4.1: Towards a globally consistent adaptation options appraisal tool, Geosci. Model Dev. Discuss., https://doi.org/10.5194/gmd-2020-151

**Authors:**

David N. Bresch (dbresch@ethz.ch), Gabriela Aznar-Siguan (Gabriela.Aznar@meteoswiss.ch)

*We thank the anonymous referee for his additional comments, which we addressed as documented below.*

*The* original comments from the referee *are listed below directly followed by our responses in blue and italic and changes to the manuscript in blue and **bold** (unless where it gets complicated or tiny, where changes are made in the manuscript only).*
* * *
I thank the authors for considering my remarks and reworking the manuscript, in particular the discussion on uncertainty. The readability has improved but the manuscript still remains a rather dense and complex piece, that requires highest attention to follow. This is further illustrated by the fact that the authors' reply requires detailed explanations that one would rather like to find in the manuscript itself.

I see no strong reason not to publish this article in its present form up to some minor comments below, but imagine that some extra hours of work on the writing could greatly benefit the reader.

*We appreciate the fact that you welcome our more in-depth discussion of (drivers of) uncertainty – thanks again, as this truly improved the manuscript. We carefully re-read the whole text and edited longer and more complex sentences once more. As there are many (small) changes, we do not list them all here, but submitted again both a clean as well as a version with track changes.*

Please see also some points below that I encountered:

27

28    Comments:

29    1. Page 3, line 11 states: "Changes in vulnerability could easily be taken into account, too, but

30    pertinent information does usually not exist. Adaptation measures are implemented through

31    modification of the impact function…", whereas the authors state on page 5, line 22: "… a set of

32    impact functions representing the exposures vulnerability…". Somehow the definition of

33    vulnerability differs throughout the manuscript. It seems that the authors aim to change

34    vulnerability through adaptation, that is implemented by modifying the impact function. Please

35    correct the manuscript accordingly.

36    *The concept of vulnerability is differently defined by different communities. We did cover this in*

37    *the first paper (Aznar-Siguan and Bresch, 2019), where we wrote "While 'vulnerability function'*

38    *is broadly used in the modeller community, we refer to it as 'impact function' to explicitly*

39    *include the option of opportunities (i.e. negative damages)." While CLIMADA does allow for*

40    *adaptation options to be implemented also through changes in hazard and/or exposure, the most*

41    *common – and frequent – use so far is indeed through modification of the impact functions.*

42    *Therefore, we amended as follows (please note that we edited the whole paragraph, i.e. we start*

43    *earlier than the mentioned sentence – please see the version with track change for all edits):*

44    *"Framing the climate adaptation challenge in terms of risk allows to treat adaptation measures*

45    *as ways to reduce natural hazard risk both today and in future.* **The open-source CLIMADA**

46    **platform (Aznar-Siguan and Bresch 2019b) integrates hazard, exposure and vulnerability to**

47    **(i) assess risk in various metrics and (ii) quantify socio-economic impacts.** *Please note that*

48    *CLIMADA does provide global coverage of major hazards beyond tropical cyclones (TC), yet we*

49    *focus* **o***n TC in the present paper for illustrative purposes. Starting from a calibrated impact*

50    *model to assess current risk , the drivers of risk are*

51    *first modified to implement the effect of changes in hazard (e.g. climate-driven) and exposure*

52    *(development-driven) over time, likely in a scenario fashion (c.f. Serrao-Neumann and Low Choy*

53    *(2018)).*

54

55    **Vulnerability is implemented in CLIMADA by means of hazard- and exposure-specific impact**

56    **functions. Past and on-going changes in vulnerability could thus easily be taken into account**

57 *in this framework, too, but pertinent information does usually not exist.*

58 *Adaptation measures are implemented* **primarily** *through modification of the impact function*

59 *(e.g. better building codes leading to lower building damages).* ***Adaptation measures which***

60 ***modify exposure (e.g. spatial planning) or hazard (e.g. dikes)  or even combinations thereof,***

61 ***could be realised in CLIMADA too, but are not part of the present illustrative application.*** *Risk*

62 *metrics both for today as well as for future years are thus calculated with and without* **each**

63  *adaptation measure.”*

64 *Throughout the text, we re-checked for the consistency of the terms vulnerability and impact*

65 *function, as e.g. properly distinguished in (Pag5, Line 24) "a set of impact functions*

66 *representing the exposures vulnerability".*

67 2. Page 2, Line 2: need -> needs

68 *Thanks, corrected.*

69 3. Page 3, line 10: , yet we focus on TC in …

70 *Thanks, too, corrected, see text above under point 1.*

**AUTHORS' RESPONSE TO REFEREE #2, SECOND ROUND**

**Research article:**

Bresch, D. N. and Aznar-Siguan, G.: CLIMADA v1.4.1: Towards a globally consistent adaptation options appraisal tool, Geosci. Model Dev. Discuss., https://doi.org/10.5194/gmd-2020-151

**Authors:**

David N. Bresch (dbresch@ethz.ch), Gabriela Aznar-Siguan (Gabriela.Aznar@meteoswiss.ch)

*We thank the anonymous referee for his additional comments, which we addressed as documented below.*

*The* original comments from the referee *are listed below directly followed by our responses in blue and italic and changes to the manuscript in blue and **bold** (unless where it gets complicated or tiny, where changes are made in the manuscript only).*
* * *
The Section 4 (Discussion and Outlook) is now becoming a very long section with long paragraphs. I suggest the authors can break it into several paragraphs in different highlighted points (such as uncertainty assessment/description).

*We carefully re-read the whole text and edited longer and more complex sentences once more. In doing so, we also broke Section 4 (Discussion and Outlook) into several paragraphs and highlighted the main points in italic font at the beginning of each such paragraph (concept of probabilistic options appraisal, assessment of uncertainties, combining g measures, development of adaptation strategies ).*

*As there are many (small) changes, we do not list them all here, but submitted again both a clean as well as a version with track changes.*

[revised manuscript text omitted]

**Commented [dnb1]:** As mentioned in the reply to reviewer #2, we highlight the key point discussed in each paragraph by setting first mention into *italic*.

this paper, the results illustrate the dependence of cost-effective solutions on social and environmental conditions in a limited area and scope of study.

While CLIMADA would lend itself to a detailed *assessment of uncertainties* in all elements of the analysis, we abstained from doing so in order to keep the case study illustrative and the figures more easily readable. Main drivers of uncertainty, beyond those in hazard, exposure, and vulnerability (Aznar-Siguan and Bresch, 2019b) for the four adaptation measures, while not quantified, can at least be qualitatively described as follows. As for preparedness, the level and scope for this study have been chosen based on general findings of previous ECA studies (Caribbean Catastrophe Risk Insurance Facility, 2010), where large differences had been found across regions, mainly stemming from barriers to implementation, not least such as lack of agency of non-owner property residents. Notwithstanding, in all cases, preparedness does lower damages and almost always at a Benefit/Cost ratio >1 on a societal level, which does not necessarily mean it being 'worth the money' for the single property owner each time. As for mangroves, differences of applicability to single islands have been mentioned above. Again, as shown in studies (Reguero et al., 2018), such nature-based solutions, while difficult to assess at great precision in terms of exact Benefit/Cost, yield ratios far above one if applied at scale. With building codes, it all depends on design - and enforcement. The latter being utterly cultural, any assessment must remain spurious, even past experience might not provide solid a guidance for present and future uptake. On the other hand, implementation is rather straightforward in CLIMADA in terms of the impact function as far as the design component is concerned, hence relative uncertainty can be limited there. Retrofit is implemented the exact same way as building codes and exposed to the same threat of enforcement. Risk transfer in contrast to measures discussed so far, being a purely monetary transaction, shows, in its assessment at least, less uncertainty. But it inherits all the underlying uncertainty of the probabilistic model as well as of the measures in terms of their risk-reducing capacity. Testing with many (sets of) parameters (Aznar-Siguan and Bresch, 2020), results regarding the effectiveness of risk transfer proved robust and, most importantly, the relative order of measures in terms of cost-effectiveness in general is very robust, too.

We show that *combining measures* of different nature, such as mangrove restoration, preparedness, building codes enforcement and retrofitting, can increase the amount of averted damage in a cost-effective way. In Anguilla restoring mangroves averts simulated expected damage of more than 40 m USD over the next 35 years in a moderate climate change scenario. This represents 6% of the scenario's expected damage and more than 4 times the cost estimated for restoration (confirming the finding of a study by the Caribbean Catastrophe Risk Insurance Facility (2010) as well as e.g. Reguero et al. (2018)). Furthermore, combining mangroves restoration together with building codes enforcement and preparedness increases the averted damage to 14% of the expected damage by 2050 with a cost of approximately one third of the overall benefit. On the other hand, in the Turks and Caicos islands enforcement of building codes results in a more cost-effective measure than mangrove restoration, with a cost ¾ of the benefit. The reason being, even if both measures avert an expected damage of approximately 220 m USD (11% of the islands expected damage), the restoration of mangroves needs to be implemented just in targeted areas to avert more damage than the invested capital. In these islands the combination of grey measures, namely

**Commented [dnb2]:** Note to the editor: As final typesetting will vary, breaks between paragraphs might look different (we did just insert an empty line here).

**Commented [dnb3]:** As mentioned in the reply to reviewer #2, we highlight the key point discussed in each paragraph by setting first mention into *italic*.

Note that we also broke into a new paragraph here, as the previous sentence in fact betters concludes the paragraph above.

**Commented [dnb4]:** As mentioned in the reply to reviewer #2, we highlight the key point discussed in each paragraph by setting first mention into *italic*.

building codes enforcement and retrofitting, with preparedness education averts 35% of the simulated expected damages in a moderate climate change scenario by 2050 with a high - but effective - cost of 97% of the benefit. In order to avert a significant fraction (more than 50%) of the expected damage over the next 35 years, risk transfer shows to be the most effective complement in all cases studied. Combining insurance – be it indemnity-based or parametric (Caribbean Catastrophe Risk Insurance Facility, 2015) – with other cost-efficient measures reduces its cost (by 50 m USD in the case of Anguilla and 250 m USD in the Turks and Caicos islands) and covers damage of high impact events which cannot be cost-effectively averted by other measures.

While the idealized case study already provides elements relevant for the *development of adaptation strategies* and the interplay of prevention, preparedness and risk transfer (c.f. Joyette et al. (2015)), further locally bespoke data would improve the accuracy and representativeness of results, starting from spatially-explicit mapping of specific exposures such as infrastructure and sectoral split. The concept could handle both indirect impacts (business interruption etc.) as well as series of consecutive events, but we did not venture into these forays for lack of data to validate the model with. Similar for other related hazards, where one could expand to torrential rain and separate wind from surge action for tropical cyclones, which would pose a major challenge as there is no separately reported damage and indirect methods would introduce further uncertainties (Strobl, 2012).

*Methodologically,* as mentioned in the introduction, Cost Benefit Analysis (CBA) and Multi-Criteria Analysis (MCA) are not mutually exclusive, and a comprehensive analysis builds on both used in conjunction with each other. While ECA studies in cities in other regions such as in El Salvador or Bangladesh (Wieneke and Bresch, 2016) did consider asset damage and impacts on people affected and lives lost in their pertinent metrics, some earlier studies also compared health impacts of reduced reservoir outflow (measured in disability adjusted life years as in Finkel (2019)) with hydropower production (measured in MWh and electricity costs) in Tanzania (Bresch and ECA working group, 2009). In order to keep the case study exemplary and simple, we did not introduce metrics other than direct damage. This bears the advantage to easily compare very different adaptation options and their damage aversion potential by reducing all to a common monetary form. But we do see eminent potential to develop both the platform and its applications in the direction towards MCA, with determination of relative weights of the multiple criteria remaining a challenge. So far, ECA analyses in general and the underlying CLIMADA platform in particular do not account for a range of critical factors such as the role of institutions, access to and ownership of resources, agency and leadership (Preston and Stafford-Smith, 2009), to name a few. While this will remain a major challenge for long, we do see potential for further development of both the ECA methodology as well as the CLIMADA platform to better serve adaptive management approaches. as they The approach can be used to support the measurement of successful implementation and hence evaluation of proposed adaptation options over time. We and therefore provide a framework for lessons learned to inform future actions in an iterative fashion to make better informed, and often incremental, decisions in the face of uncertainty (Wilby and Dessai, 2010). Such a dynamic simulation platform mitigates shortcomings of static adaptation databases (Mitchell et al., 2016) and lends itself as a basis for web-based adaptation support tools (Glaas et al., 2017). and Hence it might hence

**Commented [dnb5]:** As mentioned in the reply to reviewer #2, we highlight the key point discussed in each paragraph by setting first mention into *italic*.

**Commented [dnb6]:** As mentioned in the reply to reviewer #2, we highlight the key point discussed in each paragraph by setting first mention into *italic*.

[revised manuscript text omitted]